Local domestication of lactic acid bacteria via cassava beer fermentation

Colehour Alese M. 1 2 alese@uoregon.edu
Meadow James F. 2
Liebert Melissa A. 1
Cepon-Robins Tara J. 1
Gildner Theresa E. 1
Urlacher Samuel S. 3
Bohannan Brendan J.M. 2
Snodgrass J. Josh 1
Sugiyama Lawrence S. 1
1 Department of Anthropology, University of Oregon , Eugene, OR , USA
2 Institute of Ecology and Evolution, University of Oregon , Eugene, OR , USA
3 Department of Human Evolutionary Biology, Harvard University , Cambridge, MA , USA
Smidt Hauke
Electronic publication date: 2014 Jul 8
Publication date: 2014
Volume: 2
Electronic Location ID: e479
Received 2014 Feb 2; Accepted 2014 Jun 18
Copyright: © 2014 Colehour et al.
Copyright year: 2014
Copyright holder: Colehour et al.
License: This is an open access article distributed under the terms of the Creative Commons Attribution License, which permits unrestricted use, distribution, and reproduction in any medium, provided the original author and source are credited.
License URL: https://creativecommons.org/licenses/by/3.0/

Keywords: Fermentation, Microbial domestication, Food microbiology, High-throughput sequencing, Lactobacillus, Lactic acid bacteria, Artisanal starter culture, Alcoholic beverage, Maniot, 16S ribosomal

Funding: Graduate Research Fellowship Program of the National Science Foundation 2011102824 Wenner-Gren Foundation 8476 Sasakawa Young Leaders Fellowship Fund Department of Anthropology (UO) Institute of Ecology and Evolution (UO) Harvard University Frederick Sheldon Traveling Fellowship This study was funded by the Graduate Research Fellowship Program of the National Science Foundation, Fellow ID 2011102824 (https://www.fastlane.nsf.gov/grfp/). Other incurred project costs were supported by the Wenner-Gren Foundation (Grant number 8476; http://www.wennergren.org/programs/dissertation-fieldwork-grants), the Sasakawa Young Leaders Fellowship Fund (http://www.tokyofoundation.org/sylff/), Department of Anthropology (UO) (http://pages.uoregon.edu/anthro/), and the Institute for Ecology and Evolution (UO) (http://ie2.uoregon.edu/) and the Harvard University Frederick Sheldon Traveling Fellowship. The funders had no role in study design, data collection and analysis, decision to publish, or preparation of the manuscript.

==============================
Cassava beer, or chicha, is typically consumed daily by the indigenous Shuar people of the Ecuadorian Amazon. This traditional beverage made from cassava tuber (Manihot esculenta) is thought to improve nutritional quality and flavor while extending shelf life in a tropical climate. Bacteria responsible for chicha fermentation could be a source of microbes for the human microbiome, but little is known regarding the microbiology of chicha. We investigated bacterial community composition of chicha batches using Illumina high-throughput sequencing. Fermented chicha samples were collected from seven Shuar households in two neighboring villages in the Morona-Santiago region of Ecuador, and the composition of the bacterial communities within each chicha sample was determined by sequencing a region of the 16S ribosomal gene. Members of the genus Lactobacillus dominated all samples. Significantly greater phylogenetic similarity was observed among chicha samples taken within a village than those from different villages. Community composition varied among chicha samples, even those separated by short geographic distances, suggesting that ecological and/or evolutionary processes, including human-mediated factors, may be responsible for creating locally distinct ferments. Our results add to evidence from other fermentation systems suggesting that traditional fermentation may be a form of domestication, providing endemic beneficial inocula for consumers, but additional research is needed to identify the mechanisms and extent of microbial dispersal.

Introduction

Fermentation converts simple carbohydrates into secondary compounds, including alcohols and lactic acid, and is used by human societies worldwide as a means to improve the flavor, nutritional value, and/or preservation of food and drinks (Campbell-Platt, 1994; Van Hylckama Vlieg et al., 2011). Fermentation is mediated by a variety of microorganisms; for example, yeasts convert carbohydrates into carbon dioxide and alcohol to produce alcoholic beverages, while Lactobacillus bacteria create lactic acid, the tangy flavor characteristic of food such as sauerkraut and yogurt. Conventional fermentation utilizes commercially available bacteria or yeast for fermentation, often from a single laboratory-isolated strain. In contrast, spontaneous fermentation—sometimes called traditional or wild fermentation—exposes carbohydrates to diverse microorganisms resident in the environment to cultivate a ferment (Campbell-Platt, 1994; Scott & Sullivan, 2008; McGee, 2013). Often, diverse communities of bacteria and yeast undergo ecological succession in the fermentation vessel as the community structure changes in response to conditions created by preceding species.

In a liquid-substrate ferment of cassava (Manihot esculenta), lactic-acid bacteria (LAB) dominate throughout the process but specific composition is variable as the bacterial byproducts increase the acidity, shifting the pH from 6.5 to around 4.5 after 48 h (Abriba et al., 2012; Tetchi et al., 2012). LAB are commonly associated with nutrient rich-substrates belonging to a range of plants and animals. Sources of bacteria in a lactic-acid ferment could include inocula from the raw material (plant substrate or water), airborne dust, fermentation tools/vessel, or bacteria introduced by humans supervising the fermentation process. Site-specific spontaneous fermentation replicated over long periods of time has been considered a form of microbial domestication, especially in Saccharomyces yeast and LAB (Fay & Benavides, 2005; Suzzi, 2011; Bachmann et al., 2012). This suggests that ecological processes inherent in spontaneous fermentation, including human-mediated selection, could result in artisanal products unique to a particular region and cultural practice (e.g., Iambic ales (Spitaels et al., 2014), Old World wines (Sun et al., 2009), cheeses (Bokulich & Mills, 2013), and sourdough breads (Scheirlinck et al., 2007)). Domestication of LAB can be demonstrated if human choice (e.g., tuber variety, batch size, fermentation length, and individual flavor preferences), and adaptation to unique environmental fluctuations (e.g., temperature, pH, and disturbance) result in a unique microbial community that is consistent over time.

Chicha is a generic term for traditional fermented beverages produced by indigenous groups throughout the Amazon basin and Andes. It can be made from a variety of starchy plant crops including maize, cassava, and millet. Archaeologists have identified traces of 1,600-year-old sprouted maize chicha in 150-l clay vats in the remains of a pre-Incan civilization in Cerro-Baul, Peru, making it one of the oldest known ferments (Moseley et al., 2005). Today, indigenous Amazonian groups continue to brew chicha (also generically referred to as masato in some regions) from sweet cassava, also called manioc or yuca, a staple tuber cultivar in tropical climates. Chicha from cassava is typically a low alcohol beverage (2%–5%), with a milky consistency and somewhat sour flavor. Amongst Shuar, chicha (nijiamanch’ in Shuar) is typically prepared over a 2–3 day period. First, the roots of yuca are peeled, washed, and boiled until soft. Water is then drained off and the root mashed (traditionally on a large wooden platter but now commonly in a large metal pot) with a dedicated pestle, while the brewer masticates pieces of the yuca and periodically spits into the mash. Recipes vary according to the brewer’s taste. For example, different locally identified varieties of yuca can be mixed together, or yam (Dioscorea sp.) is sometimes added. The finished mash is placed in a designated vessel to ferment for 1–3 days, depending on preference for sweet (slightly to unfermented) to sour (very fermented) chicha. While traditional fermentation vessels were ceramic, today 3–5 gallon plastic buckets are typically used. Water is then mixed with the fermented mash just prior to serving, with consistency based on desired water to mash content.

First domesticated in neo-tropical lowland South America 8,000–10,000 years ago, cassava continues to be a dietary staple across this region for many Amazonian forager-horticulturalist groups (Piperno, 2011). Fermented cassava beer remains a key component of the diet for many, with fermentation improving bioavailability and synthesis of essential vitamins and minerals (zinc, calcium, iron and magnesium) that may otherwise be lacking (Boonnop et al., 2009; Ahaotu, Ogueke & Owuamanam, 2011; Dilworth, Brown & Asemota, 2013). This is particularly important since chronic nutritional stress among indigenous groups can stunt growth (Blackwell et al., 2009; Piperata et al., 2011). Furthermore, fermentation facilitates decomposition of organic toxins such as naturally occurring cyanides in yuca that cause weakness, hypothyroidism, and paralysis (Lei, Amoa-Awua & Brimer, 1999).

Despite the importance and widespread consumption of chicha, no studies to date have characterized the microbial community present in chicha using modern culture-independent techniques. Several groups of lactic-acid bacteria including Lactobacillus spp., have been detected in cassava ferments using culture-dependent methods (Axelsson, 2004; Santos et al., 2012). Other research on similar cassava ferments documented the sharpest decrease in sugar content occurring between 24 and 48 h, corresponding with increasing mass of LAB reaching peak CFU/g at 72 h after fermentation is initiated (Tetchi et al., 2012). In culture-dependent sugar metabolism profiling experiments, 90% of LAB strains isolated from a cassava ferment digest glucose, gluconate, maltose, melibiose, raffinose and sucrose, while less than 10% of strains fermented esculin, cellobiose, glycerol, mannitol, melizitose and rhamnose (Kostinek et al., 2005). Isolates from cassava fermentation also exhibited high levels of hydrogen peroxide and bacteriocin production, antimicrobial agents produced by LAB antagonistic toward closely related strains (Kostinek et al., 2007) and toward fungal species (Adebayo & Aderiye, 2010).

Some species of Lactobacillus are considered to be beneficial to human health, given their ability to compete with pathogens, stimulate mucus production, and bind to the lining of the intestinal tract (Kravtsov et al., 2008; Lebeer, Vanderleyden & De Keersmaecker, 2008; Turpin et al., 2012). Certain strains also improve the uptake of nutrients by enhancing mineral absorption, promoting host growth factors, and degrading antinutrients (e.g., digestion inhibitors synthesized as a plant’s self-defense against herbivores) (Turpin et al., 2010). Commercially-isolated Lactobacillus strains are commonly added to pasteurized dairy products such as yogurt or sold in capsule form as an increasingly popular solution for an array of common health problems, including irritable bowel syndrome and other conditions related to chronic inflammation of the intestinal tract (Allgeyer, Miller & Lee, 2010; Ranadheera, Baines & Adams, 2010; Yang & Sheu, 2012). In contrast, a few species of LAB are also known producers of biogenic amines in vegetable ferments but these are only toxic in high concentrations when they accumulate in ferments lasting several weeks (Halász et al., 1994). Traditional amylaceous (starch-based) ferments such as chicha may contain novel strains of Lactobacillus that might be a source of microbes in indigenous populations (Aro, 2008; Chelule, Mokoena & Gqaleni, 2010; Chelule et al., 2010; Van Hylckama Vlieg et al., 2011). Much is known about the health benefits of certain strains but these cannot be generalized across all species of LAB since specific characteristics can vary dramatically (Makarova et al., 2006; Turpin et al., 2010).

Just as in any ecosystem, bacterial communities in fermented foods are shaped by a variety of ecological processes, including environmental selection and dispersal, that select for a subset of potential inhabitants from a metacommunity (a set of communities linked by dispersal of multiple, interacting species). Lactic-acid bacteria in spontaneous ferments have the unique ability to survive both nutrient saturation and starvation, suggesting that some Lactobacilli are well adapted to fermentation of food both inside and outside the gastrointestinal tract (Ganesan, Dobrowolski & Weimer, 2006; Suzzi, 2011; Van Hylckama Vlieg et al., 2011). Recent research shows spatial diversification of bacteria and yeast populating artisan cheese cultures that correlate with those of specific surfaces in the processing facility (Bokulich & Mills, 2013), hinting at the possibility that microbial communities undergo geographic divergence in human-mediated ecosystems. However, it is unclear whether bacterial community composition in a small-batch spontaneous ferment shows spatial diversification that would suggest incidental domestication.

To examine geographic variation in spontaneous ferments, we collaborated with an indigenous group of Shuar (as part of the Shuar Health and Life History Project [ http://www.bonesandbehavior.org/shuar/]) engaged in a forager-horticulturalist lifestyle in the remote Cross-Cutucú region of the Ecuadorian Amazon. We assessed phylogenetic similarity of bacterial communities in chicha batches from households in two villages to determine whether: (1) a ferment from one house was significantly different than ferments from neighboring households in the same village, and; (2) ferments from different households in the same village were more similar to each other than to batches from households in a different village.

Materials & Methods

Population and location

All samples were collected in the Cross-Cutucú region of the neo-tropical lowlands of southeastern Ecuador, which lies east of the Cutucú Mountains along the Morona river drainage. This region has an annual rainfall of more than 4,000 mm (158 inches) and average daytime temperatures of 29 °C (85 °F) (Kricher, 1999).

The Shuar of this region are an indigenous forager-horticulturalist group who live primarily in small riverine villages. They are a natural fertility population and traditionally lived in scattered matrilocal household clusters in traditional thatch-wall, earthen-floor houses (Karsten, 1935; Descola, 1996; Rubenstein, 2001), although plank houses with tin roofs are becoming more common. Present day subsistence remains based on horticulture, fishing, hunting and gathering, yet they are currently experiencing increasingly rapid infrastructure development and market integration as a result of regional economic development (Blackwell et al., 2009; Liebert et al., 2013). However the villages in the present study subsist with limited daily access to markets or exposure to economic development. Rates of infectious disease remain high throughout this population, accounting for 15% adult mortality in 2008 (World Health Organization, 2011; McDade et al., 2012). Further, Cepon-Robins et al. (2013) reported that 65% of the population in this particular region is infected with parasitic worms, with even higher prevalence among children. Stunting among children is a common public health concern, and is relevant to ongoing studies investigating metabolic health in the context of economic transitioning populations (Santos & Coimbra, 2003; Foster et al., 2005; Orellana et al., 2009; Blackwell et al., 2009; Liebert et al., 2013) making nutrition-related health research a high priority in this region.

Sample collection

We collected samples in two villages in the Cross-Cutucú region of Morona Santiago, Ecuador. Village 1 (V1; pop. 50) is located approximately one to four hours by motorized canoe (depending on water levels) from the nearest port with road access. A nearby spring located upstream from the village provides water for bathing and cooking. Village 2 (V2; pop. 400) is located twenty-minutes by foot from V1 (including a bridgeless river crossing). Water is pumped from the river to a reserve that flows through pipes to some houses. In both villages, each household has their own chicha ferment, containing brews that are commonly maintained by the resident women. New batches are produced every 3–5 days or as needed.

We collected 2 mL of mature chicha from five households in V1 and two households in V2, during August 2012 (sample volume was limited due to limited freezer space on site). Over a period of two weeks, we collected samples from each of these seven ferments up to three times, each representing independent batches (with a shared starter culture). Fermentation maturity of samples was confirmed with litmus paper ensuring a pH range between 4.0 and 4.5 (Luedeking & Piret, 1959; Santos et al., 2012). We sampled 300 mL of spring water (concentrated on a 0.45 µm pore, cellulose acetate filter) that residents in V1 use to prepare chicha. We were unable to collect water from V2 due to equipment malfunction. All samples were immediately frozen (−20 °C) before being transported and stored at the University of Oregon until they were processed. All samples were examined under a light microscope for evidence of helminth eggs or macrophages.

Ethics statement

This study was conducted in Shuar villages located within Canton Tiwintza, Morona Santiago, Ecuador. Research for the Shuar Health and Life History project was authorized in a letter provided by the Federación Interprovincial de Centros Shuar (FICSH). No human data was gathered as part of this project, and the bacterial data gathered was purged of human mitochondrial sequences by removing all sequences classified within the Order Rickettsiales before archiving. Genetic material resulting from this research will never be used for human DNA research or commercial cell-line patenting.

Bacterial DNA extraction and sequencing

Whole genomic DNA was extracted from all samples using MO BIO Power Plant Pro kit including phenol separation solution step (MO BIO Laboratories, Carlsbad, CA) and amplified on the V4 region of the 16S rRNA (F515/R806 primer combination: 5′-GTGCCAGCMGCCGCGGTAA-3′, 5′-TACNVGGGTATCTAATCC-3′) (Caporaso et al., 2010). DNA amplifications were performed in triplicate and pooled prior to sequencing. The reverse primer included a 12 bp Golay barcode for demultiplexing in downstream analysis. PCR conditions followed Caporaso et al. (2010). Amplicons were purified using gel electrophoresis and the MO BIO UltraClean GelSpin DNA extraction kit. Equal amounts of purified amplicons from each sample were pooled and sent to the Dana Farber Cancer Institute Molecular Biology Core Facility (http://www.dana-farber.org), to be sequenced on the Illumina MiSeq platform using a paired-end 250 bp protocol. All sequences have been deposited in the MG-RAST archive under accession numbers 4545634.3–4545652.3.

Sequence processing and statistical analysis

Sequence processing was conducted in QIIME (Caporaso et al., 2010) using MacQiime (version 1.6.0, http://www.wernerlab.org/software/macqiime; QIIME, RRID:OMICS_01521). Quality filtered forward reads (Phred score > 20; 250 bp) were binned with barcodes corresponding to the respective sample IDs. Operational taxonomic units (OTUs) were assigned at 99% genetic similarity. Representative OTU sequences were aligned to the Greengenes database (October 2012 version) and assigned taxonomic nomenclature with an RDP classifier. We rarified all samples to 19,000 sequences for even sampling depth; two samples significantly below that threshold were omitted from further analysis.

We also conducted a manual BLAST search against the NCBI 16S isolate database for the top 10 OTUs for exploratory purposes (NCBI BLAST, RRID:nlx_84530). We included these species-level NCBI database matches for visualization and reporting. Results from OTU clustering matched to the Greengenes database using an RDP classifier within the QIIME pipeline were used for all analysis purposes.

Community similarity was calculated in two different ways: with the phylogeny-based UniFrac metric (unweighted), and by calculating the number of OTUs shared between samples. Unweighted UniFrac uses phylogeny-based branch lengths generated from our rarified dataset, comparing their fractions between samples to quantify dissimilarity between communities without regard for species abundance. We determined if differences were significant using PERMANOVA (Adonis method) in QIIME. Using the UniFrac matrix, we generated a PCoA plot in QIIME and included it as Fig. 4. We then used the QIIME generated OTU table to conduct a one-way ANOVA in SPSS (SPSS, RRID:rid_000042, version 20.0.0) to investigate differences in the number of shared OTUs across households and across villages (Fig. 3).

Results

We generated a total of 1,055,214 barcoded sequences 249 base pairs in length. Sequences were quality filtered and rarefied to 19,000 OTUs per sample. The nineteen samples used for analysis represent one to three chicha batches from each of 7 different households (five from Village 1 and two from Village 2). Clustering of OTUs revealed that members of the genus Lactobacillus dominated the bacterial communities in all samples (Fig. 1). Of the ten most abundant OTUs across samples, nine were Lactobacillus; the other was an Acetobacter. These 10 OTUs each represented >1% of each sample, collectively accounting for 71% of the sequences in all samples. The top two most abundant species predicted through a BLAST search on the NCBI database, L. acidophilus and L. reuteri, account for 51% of the entire dataset (Fig. 2). Two of the most abundant Lactobacillus OTUs were less than 98% similar to existing isolates in the NCBI database, potentially suggesting the presence of previously undescribed taxa, though we were not able to assess this with short 16S sequence reads. Figure 2 provides a descriptive table of the most abundant species, as predicted by a BLAST search in the NCBI 16S isolate database, as well as an isolate source habitat for each.

Figure 1 All chicha samples were dominated by Lactobacillus spp.

OTU predicted identities are from a manual BLAST search against the NCBI 16S isolate database. We found ten OTUs with greater than 1% relative abundance. This barplot shows the relative abundance (%) of ten most common bacteria isolates for each sample. Up to three independent batches were sampled for each house (e.g., H1a–H1c). Eleven samples were analyzed from Village 1 and three from Village 2.

Figure 2 All chicha samples were dominated by Lactobacillus spp.

We report the NCBI accession number and the environment where isolates with the closest match to our OTUs were identified. Many of these Lactobacillus species were also identified in the human intestinal tract. Lactobacillus acidophilus and Lactobaccillus reuteri make up 51% of all OTUs in our rarified chicha samples. Collectively, these ten OTUs account for 71% cumulative abundance for all samples. “Unknown” samples had no match above 97% similarity to an existing NCBI submission.

The bacterial communities detected in water samples had higher phylum level diversity than chicha (Supplemental Information; 127,558 OTUs per sample). Whereas Lactobacillaceae dominated chicha, Delftia acidovorans (NC_010002), a member of the Comamonadaceae first isolated from a sewage treatment plant (Schleheck et al., 2004), was the most abundant OTU encountered in water (Supplemental Information; 18.4%). The bacteria most commonly shared between water and chicha were species within the genus Acetobacter. This clade oxidizes alcohol and sugar to create acetic acid and some members of this genus have been found in traditional balsamic vinegar production (Gullo, De Vero & Giudici, 2009). Overall, community composition of chicha was very different from water, indicating the bacterial population is driven by more than just the water source.

The bacterial community in an unfermented sample is dominated by Bacillus (64%) and chloroplast (22%) clusters (Fig. 1), presumably from the plant’s genetic material before it is degraded during boiling and fermentation. We opted to leave chloroplast sequences in the dataset since levels were negligible in mature ferments (<1%), and the abundance in the unfermented sample is potentially informative (Supplemental Information). Low levels of Lactobacillus are present in the unfermented sample (1.5%) but the increase in Lactobacillus sequences in the finished product is accompanied by a dramatic reduction in the OTUs found in the unfermented sample.

Since this population is known to carry a high parasite burden (Cepon-Robins et al., 2013), we examined our samples for evidence of viable parasites. No helminth eggs or macrophages were detected in the samples. However, the samples underwent two freeze-thaw cycles before microscopic examination, which is known to reduce visibility of parasites. Additional research is needed before we can present any evidence that fermentation affects the transmission of parasites.

Bacterial communities in chicha were significantly different across the two villages (Fig. 4; F1,12 = 1.11, R2 = 0.08; p = 0.038; from PERMANOVA on unweighted UniFrac dissimilarity matrix), but they were not significantly different across households within a village (F5,8 = 0.38, p = 0.73). Water samples were significantly different from the chicha samples (Supplemental Information: F1,17 = 8.25, p = 0.005). More OTUs were shared between households within a village than across villages (Fig. 3: MDifferentV illage = 7.44, MSameV illage = 8.31, F1,89 = 4.11, p = 0.046), but batches from the same household did not have more OTUs in common than they did with other batches from houses in the same village (MDifferentHouse = 7.93, MSameHouse = 8.27, F1,89 = 0.28, p = 0.60).

Figure 3 Chicha from the same village contain more shared OTUs.

We counted shared OTUs between every possible combination of chicha samples. The average number of matching OTUs between samples that are paired within the same village was significantly higher than those paired from different villages. We considered chicha samples from different batches within the same house as independent since there was no significant difference by house. They are grouped together in this figure for visualization purposes.

Figure 4 Principle coordinate analysis by village.

A PERMANOVA (Adonis) on an unweighted UniFrac dissimilarity matrix shows significant differences in bacterial populations when grouped by village (F1,12 = 1.11, R2 = 0.08; p = 0.038). While several of the samples from Village 1 are closely clustered those from Village 2 are relatively more spread out, suggesting higher variance could partially explain the significant results. More samples, evenly distributed between villages, will likely be needed to better understand the extent and pattern of site-specific differences.

Discussion

Humans continuously and intimately interact with microorganisms. This study demonstrates that spontaneous, or traditional, fermentation promotes a diversity of microorganisms, including some Lactobacillus strains that may potentially interact with human and environmental microbes during production and consumption. Spontaneous fermentation and consumption of its product can be a microbial exchange between the environment and the human microbiome that is mediated by human behavior, abiotic factors, and random chance (Fig. 5). The microbial community of chicha could be initiated from a variety of sources including saliva added to each new batch, tools and vessels that may contain remnants or bacterial residue from a previous ferment, the water added to thin the cassava mash, the substrate of the raw material, or the household and airborne environment. In turn, ferments by lactic acid bacteria are consumed and become a potential source of microbes for the human body. While not all LAB confer benefits, many Lactobacilli have been positively associated with human health. (Dethlefsen, McFall-Ngai & Relman, 2007; Costello et al., 2009; Spor, Koren & Ley, 2011; Human Microbiome Project Consortium, 2012; Linnenbrink et al., 2013). All of the numerically dominant OTUs we detected in chicha were related to Lactobacillus species that have also been reported in the human oral and fecal microbiome (Dewhirst et al., 2010). Our conclusions are based on short reads so a much more detailed study is necessary to determine if any of the Lactobacillus taxa we detected might confer health benefits or even successfully assimilate into the human microbiome.

Figure 5 Conceptual model of microbial exchange between human cultivators and a locally distinct ferment.

Spontaneous fermentation is an ecological phenomenon driven by distance-limited dispersal, human- mediated selection, and stochastic succession that may explain geographically diversified lactic acid fermentation. Over many generations, this process can be considered a process of microbial domestication if microbe assemblages are consistently distinct.

Lactic-acid bacteria are found in association with nutrient rich environments on animals and plants. While some strains produce biogenic amines that can be detrimental to human health (Halász et al., 1994), other research highlights positive effects of consuming LAB in moderate amounts. In the human intestinal tract, high rates of adhesion to the mucus membrane allow for direct interface with the human intestine, and have been shown to protect against pathogens, modulate immune response, and promote mucus secretions to soothe the intestinal lining. In addition, lactic-acid bacteria provide digestion assistance, improving vitamin and mineral bioavailability while degrading antinutrients and other phytotoxins such as cyanide (Campbell-Platt, 1994; Westby, Reilly & Bainbridge, 1997; Aro, 2008; Chelule, Mokoena & Gqaleni, 2010; Turpin et al., 2010).

To better understand the spatial and household variability in microbial community composition in spontaneous ferments, we were interested in knowing if bacterial composition showed phylogenetic divergence over geographic space. We observed that the bacterial communities in chicha were more similar within a village than between villages (p = 0.038). This variation could result from a combination of mechanisms, including distance-limited dispersal, stochastic succession (including horizontal gene-transfer), and human-mediated selection. Since our sample size is modest, we recognize that the significant difference we found could be driven by one particular sample (Fig. 1) but our Adonis results are supported by our ANOVA analysis showing significant within-village OTU overlaps, strengthening evidence for higher rates of bacterial similarity within a village.

We observed that chicha is generously shared with neighbors within a village so it is not surprising that we did not see significant phylogenetic dissimilarity at the household level. Dispersal between chicha ferments could occur if starter cultures are mixed or if a brewer contributes saliva to her neighbor’s chicha. Based on ethnographic evidence, oral microbiome swapping is actually more likely to happen during consumption, when the drinking cup, pilchis, are passed around a social gathering and dipped repeatedly into the fermentation vessel after each individual takes a drink. Distance and geographic barriers (e.g., a bridgeless river in our case) limit social interaction and subsequently the opportunity for dispersal between the two villages, which may partially explain the observed variation in the bacterial community (Bokulich & Mills, 2013; Linnenbrink et al., 2013).

Each new chicha batch represents a unique opportunity for succession, which could be contingent on the order and frequency of species arrival. In addition, competition between microbes, abiotic conditions, rate of horizontal gene transfer, and random chance could all shape the communities within each chicha vessel. The water source used in the fermentation vessel (V1: hauled in vessels from a spring; V2: piped to outdoor spigots near houses from a reservoir) may represent a source of either facilitative or competing microbes that could influence the final composition of the ferment. Since households within a village rely primarily on the same water source, this could help explain why chicha is more similar within a village but not within an individual household. It is possible that soil differences between gardens or village areas could yield variation in the plant-associated bacteria present on the raw material, but this seems unlikely since the raw material is peeled and boiled in preparation for fermentation, although contamination of pre-boiled product via tools and skin is possible.

Differences in ferment cultivation practice between the two villages may also contribute to variation (human-mediated selection). Expressions among the Shuar such as “the prettiest girls makes the best chicha” suggest individuality, personal preferences, or oral hygiene could also play a role in the cultivation of this ferment. The addition of saliva could be another source of variance, particularly if genetic or lifestyle differences contribute to distinct oral microbiome communities. Finally, any bacteria leftover in the fermentation vessel or on the tools used to make the mash may act as a starter culture for each new batch. Although beyond the scope of this article, it is tempting to speculate that distinct bacterial communities may be maintained over time, and that the combination of these factors suggest an example of microbial domestication.

These three processes, human-mediated selection, distance-limited dispersal, and stochastic variance help explain the cross-sectional bacterial community variation we observed in chicha. If site-specific bacteria communities are consistent over time this would indicate that the ferment may be a player in a co-evolutionary relationship with human and environmental microbes (Fig. 5), but future work with higher sample size is necessary to explore these ideas in greater depth.

Supplemental Information

Figure S1 Water has higher OTU diversity than chicha

Wat1 – water sampled from village 1 H1 – houses 1–6, time points a-cV2H1a – Village 2; houses 1–2, time points a-b Unf1 – Unfermented sample from village 2 Note: Samples H6a and H5a were excluded during analysis due to insufficient OTU count. See “Supplemental Legend” file for information on OTU taxonomic classification.

Click here for additional data file.

Supplemental Information 2 Legend for Supplemental Figure 1

Genus level OTU abundances for chicha, water, and unfermented sample. Sort data by column to view data on OTU abundance by sample. Sample ID Legend can be found on second sheet of the file (“Sample Legend”).

Click here for additional data file.

We thank the Shuar community members for their collaboration and hospitality. Lab work and analysis were completed in collaboration with various members in Dr. Brendan Bohannan and Dr. Jessica Green’s labs in the Institute of Ecology and Evolution, University of Oregon (UO) including Lucas Nebert, Ashley Bateman, Adam Altrichter, Ann Womack, and Adam Burns. We would also like to acknowledge two anonymous reviewers for providing us with thoughtful feedback on a previous version of this manuscript.

Additional Information and Declarations

Competing Interests

Author Contributions

Field Study Permissions

DNA Deposition

The authors declare there are no competing interests.

Alese M. Colehour conceived and designed the experiments, performed the experiments, analyzed the data, contributed reagents/materials/analysis tools, wrote the paper, prepared figures and/or tables, reviewed drafts of the paper.

James F. Meadow contributed reagents/materials/analysis tools, prepared figures and/or tables, reviewed drafts of the paper.

Melissa A. Liebert, Tara J. Cepon-Robins, Theresa E. Gildner and Samuel S. Urlacher reviewed drafts of the paper, field support.

Brendan J.M. Bohannan contributed reagents/materials/analysis tools, reviewed drafts of the paper.

J. Josh Snodgrass reviewed drafts of the paper, advising and editing.

Lawrence S. Sugiyama reviewed drafts of the paper, advising, editing, and field support.

The following information was supplied relating to field study approvals (i.e., approving body and any reference numbers):

Research for the Shuar Health and Life History project was authorized by letter issued by the Federación Interprovincial de Centros Shuar (FICSH).

The following information was supplied regarding the deposition of DNA sequences:

MG-RAST 4545634.3–4545652.3

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
