# Peer review of "Local domestication of lactic acid bacteria via cassava beer fermentation"

_PeerJ, doi:10.7717/peerj.479_

## Round 0.1 · original submission · Major Revisions

As you will see from the detailed comments provided by two expert reviewers, your study is of interest with respect to the microbial communities associated with chicha fermentation. Nevertheless, both reviewers identified a number of major shortcomings, such as those with respect to unsupported health claims, incomplete presentation of data (community composition), complete lack of analysis of the fungal (yeast) component, and lack of data from raw materials as potential source of inoculum.

Reviewer 1 ·

Basic reporting

As far as i can this article conforms to Peer J policies and templates. It contains sufficient intro/background and the figures are relevant.

Experimental design

I have some concerns on this article. First and foremost the authors left out any fungal (yeast) analysis. It is unclear how much contribution to the fascinating chicha fermentation fungal (yeast) populations provide. Methods for fungal NGS profiling are well developed (see their cited papers by Bokulich). At the very least the authors should do QPCR on all bacterial and all fungi just to get an idea of the contribution of the fungal population. If it is a significant percentage, then the fungal populations should be profiled. If not, then not.

A key missing element is to sample the cassava or, perhaps more importantly, the beginning of the fermentation. The authors sampled the water however it is far more likely that additional microbes (particularly the fungal populations) enter this fermentation from the substrate as would be the case for Kimchi. This significantly hampers this work as the assumed source of all the LAB are inferred from human (overtly and I would argue speculatively argued in Figure 3), this certainly might not be so. It is likely to be true given the species IDed, however not certain given no substrate analysis was done.

No end product analysis was done on these samples. What was the final ethanol concentration (which would suggest significant yeast involvement)? what about acetic, lactic acids? Is that known for these chicha fermentations?

Validity of the findings

The methods appear valid. There is far too much speculation on the nature of the microbe found, particularly in Figure 3. No proof of domestication nor co-evolution has been demonstrated at all.

There also is far too much speculation on a purported link to probiotics, that angle should be softened or removed entirely.

Additional comments

This is fascinating work on an absolutely fascinating indigenous fermentation. I have several concerns. I find the assumption of a sole human contributor to the chicha fermentation microbiota likely, but unproven, given the substrate wasn't sampled pre fermentation to assess plant-borne bacteria and yeast.

Also I believe the relevance to "probiotics" is not supported here. To claim that any lactic acid bacteria showing up in an indigenous fermentation, even if they are of the same species of common commercial probiotics is unfounded and speculative. Moreover it lends to the broadening of the probiotic definition onto anything fermented. The whole probiotics angle of this work needs to be softened if not omitted. This is an interesting fermentation without that overt speculation.

Minor concerns:

ln49. typical wild ferment is to vague? which one? Cacao? Kimchi, Sauerkraut? this sentence is not correct as Leuconostoc do not start all wild ferments. need to be specific here. Authors need to find a fermentation roughly similar and discuss that. Cassava fermentations have been previously studied.

ln 52. lacto-fermentation is not really used. fermentation by lactic acid bacteria is

ln 86. The statement “living ferments contain viable probiotics” is not true. The use of the term “probiotic” to cover any microbe in a ferment is simply wrong. That some species of commercial probiotics are found in indigenous fermentations does not make them probiotics per se. (see comment above on overuse of probiotic aspect to this ms)

ln 173. Were the samples really frozen at -20 immediately while on site? Really? Not put in on ice and transported?

Ln 269. Brew mistress is likely not to be understood. Please define

Ln 301-3. Again unnecessary (and in my view incorrect) over emphasis on health benefits.


Figure 1 The fact that that strain which most closely matched the short 16S sequence came from a defined probiotic strain does not confirm that the L. acidiphilus or L. reuteri found in that fermentation (which should to be validated by a second method such as tareted QPCR) is a probiotic. This is a misconception by the authors. Saccharomyces cerevisiae is often isolated from the human intestinal tract, that doesn’t make it a probiotic.

Figure 3. I think Figure 3 is too speculative. No analysis of the input microbiota on the substrate, nor analysis of the fungal populations have been provided. No detailed genomic analysis of the lactobacilli strains involved has been carried out, ergo no proof of domestication or co-evolution has been proven. This should be removed completely.

Reviewer 2 ·

Basic reporting

No comments

Experimental design

See general comments

Validity of the findings

See general comments

Additional comments

This manuscript describes the investigation of chicha fermentations from two villages in Ecuador, demonstrating regional variation in these fermentations. The manuscript presents a fascinating narrative of how chicha fits in the lifestyle of the local people and I really like the goals of this study. However, I have several major concerns, namely with some of the broad-stroke conclusions made (especially regarding beneficial health properties) that do not reflect the results or methodology of the study. I think this study is a great addition to the food/fermentation microbiology literature, but think the health claims come off as rather heavy-handed, when this study really has not been executed to address the potential health properties of chicha.

My foremost concern is that the microbial consituents of these ferments are linked to health throughout, whereas the methodology and design do not remotely handle health properties. In the abstract, in the introduction, in the discussion the health benefits of lactic acid bacteria are described and we are led to believe that the lactics detected in chicha must therefore be beneficial to human health. There are several issues with this:
1) Beneficial properties have been demonstrated only in specific strains of lactics. Not every species or strain will automatically have beneficial properties. In fact, many in foods can be detrimental to human health, e.g., biogenic-amine producing lactic acid bacteria.
2) The methodology used cannot confidently identify species (see below), and thus no comparison should be made between the putative species detected and the probiotic characteristics of other strains in this same species. This is a large leap to conclusions.
3) None of the approaches used in this study relate in any way to human health properties, so even speculation as to the possible health benefits of lactics in these ferments is inappropriate in my opinion. This is the equivalent of automatically conferring probiotic properties to other foods simply based on the presence of lactic acid bacteria. Lactics are found in abundance in many different fermentations, almost all fermented foods in fact (traditional fermentations, at least), and are frequently spoilage agents. Does wine automatically become probiotic, simply because it has a high population of lactics active during the fermentation?
All said, I agree that the lactics in these fermentations very well may have beneficial properties when consumed, but it is a very large jump to conclusions that any lactic may be probiotic (especially when the ID is dubious) and health discussions are not relevant in the context of the evidence provided.

Second, there seems to be an assumption (e.g., in Fig 3) that the dominant lactics are human-derived (with the implication that they are also beneficial following consumption), leading to a long-term co-evolution of microbial constituents in these fermentations with their human creators. However, this is not supported by the evidence provided. Lactics are common on plant material as well as in foods and host-associated environments, so could conceivably come from many sources. Sampling the raw materials, built environment, and saliva would have helped address this question (as would strain profiling techniques). Lacking evidence to the contrary, I would argue that the fermentation vessel and raw materials are the most likely sources of these lactics. Following the same evolutionary arguement presented in this work, wouldn't microbes adapted to fermentation conditions (low pH, alcoholic) be more fit to re-inoculate than those adapted for human saliva (higher pH, non-alcoholic, host interface)?

Inappropriate methods are used for species-level identification of these sequences. BLAST chooses the top hit but does not consider any confidence criteria to ensure a degree of reliability. With 250bp sequences of V4 16S rRNA, species-level identification is not reliable. A method like RDP classifier uses bootstrapping to determine the confidence of assignment to a given taxon, providing a more reliable assignment (which in the case of 250 bp V4 16S rRNA reads will probably be family- or genus-level for Lactobacillus).
Therefore, many of the assumptions made in this manuscript are inappropriate, given that BLAST identifications of short amplicons are given in evidence. A more reliable classification algorithm (e.g., RDP, UCLUST, MOTHUR) should be used to correctly identify these sequences.

Furthermore, only incomplete results are actually shown. The cumulative relative abundance across all samples is shown for the top 10 most abundant OTUs. However, this does not present the community composition of individual samples and the cumulative abundance amounts to 71%, so a very small portion of the actual data is shown (and no data for water samples are shown). It would be better to show stackplots for each individual sample, showing the complete set of the most abundant OTUs for each sample.

Overall, the introduction is somewhat under-referenced, discussion of probiotic properties needs to be limited to the discussion, results need to be re-worked with an appropriate taxonomy classifier, more complete data should be shown, and the discussion is overly speculative given the data presented here.


Minor concerns:

ln 25-27 - As the probiotic value of these lactobacilli is speculative, I think this should be removed from the abstract and left for the discussion only.

ln 45-58 - A large body of prior research exists studying the microbial succession of traditional, wild fermentations (beer, wine, cheese, sauerkraut, pickles, etc). It would be worthwhile to reference these here, and much more useful to interested readers to consider these examples than the books/reviews referenced in the previous sentence.

ln 49+ - "Typical" of what? You cite kimchi fermentations, but no single fermentation can be said to be typical, as substrate, environment, and human factors drive the wide range of diversity seen in different fermentations. It may be more useful to root this discussion of general fermentation trends in the context of fermentations more similar to chicha, e.g., other alcoholic fermentations (like some traditional African grain fermentations, which have a similar process to chicha).

ln 52-57 - Your references document domestication events in Saccharomyces and Arabidopsis. I recommend referencing these directly to present the case for microbial domestication. The current wording is misleading and suggests that these studies demonstrate lactic acid bacteria domestication in repeat fermentations. To my knowledge, such studies haven't been performed previously.

ln 57-59 - Should reference the regional microbial processes involved in the foods listed.

ln 85-97 - The Saulnier reference does not support this statement. It mentions fermented milk beverages once as a delivery food for probiotics but is not an appropriate references as it discusses mechanisms of probiosis and prebiosis within the human body and discusses functional foods -- not fermentations -- as delivery foods. This statement is a broad and vague generalization and should be removed.

ln 207 - Please provide details of the taxonomic classification method used.

ln 226-228 - Possibly. PCR/Sequencing error could also result in divergence from reference sequences.

ln 239-247 - Some visual representation of the beta-diversity between samples may be useful here (e.g., Bray-Curtis PCoA plot). It is important to be able to visually observe the separation of data points and their relation to community structures (e.g., using a taxa biplot or an accompanying stackplot of different taxa), especially as between-village P values are very high.

Also, please provide the R value for the PERMANOVA tests.

ln 256-259 - Do these studies directly support this statement that mature lacto-ferments are sources of beneficial microbes? I cannot access all of these directly but this seems to be broad extrapolation. It is true that Chicha intuitively contains human-derived microbes (from saliva) but there is no evidence that (a) human-derived microbes are numerically dominant in the fermentation or (b) that they are automatically beneficial. It would be interesting and important to collect saliva samples from the brewers and swabs of the fermentation vats and raw materials, which are the other intuitive sources of microbes in the fermentation. It is also inappropriate to speculate that these are beneficial microbes without some more supporting evidence. Just because the numerically dominant lactobacilli have been reported in humans previously is invalid as: (a) BLAST is an inappropriate algorithm for species-level identification based on such short amplicons (V4 16S rRNA) and (b) just because these species are found in humans does not mean that they are not beneficial; resident pathogens are also chronically present in some individuals but their beneficial properties are doubtful.

ln 282-284 - what about the precise substrates in each brew? The same harvest of cassava, shared recipes involving adjunct substrates, etc? As the water samples bore little resemblance to the chicha, I would consider this a minor source. Other shared substrates are a more likely source of the within-village similarity.

ln 295-298 - what materials are used in the fermentation vessels? Do these wear down/break with time (e.g., wood)? Or are they vessels that are inherited and essentially indestructible (e.g., stone cauldron). This is fundamental to understanding the inter-village variability, as well as the longitudinal duration of microbial transfer/co-evolution across batches.

ln 303-308 - SOME strains of lactics, not necessarily all, and please provide references for these statements.

Fig 1 - NCBI blast searches are inappropriate for confident species-level ID of 250 bp amplicons. RDP classifier or UCLUST classifier would both be more reliable for identification of these sequences. Likewise, presenting the source environment of these BLAST hits and whether any strain within this species is used as a probiotic is completely inappropriate, as it is a large jump to conclusions (a) that this strain must have the same properties (if it is even actually a member of the same species, which is dubious with BLAST classification) (b) that use as a commercial probiotic automatically implies it has proven benefits, which is not necessarily true (probiotic status is difficult to prove, and commercial strains are not necessarily the best by any criteria).

It would be more informative and reliable to provide assignments from RDP or UCLUST and present stack plots of the taxa detected in all samples, rather than relative abundance across the entire dataset.

---

## Round 0.2 · Minor Revisions

As you will see, we found that your revised manuscript has largely improved as compared to the original submission, however, still some issues remain to be addressed, especially related to species-level assignments made by BLAST analyses, and over-interpretation & assumptions regarding functional properties not experimentally backed-up but rather made based on properties of related strains.

Reviewer 2 ·

Basic reporting

see below

Experimental design

see below

Validity of the findings

see below

Additional comments

The revised manuscript has eased many of my prior concerns, especially much of the language regarding human health benefits derived from chicha consumption. There are a couple passages, noted below, that I still find overly speculative given the nature of the experiments performed. Additionally, the use of BLAST for species-level identification is still highly problematic and has not been fully addressed; it has been noted in the methods section, but it still stands that BLAST assignments are being used to drive the conclusions of this experiment. This is not an appropriate method and thus some of the conclusions drawn are overly speculative, as mentioned below.

Abstract — The revised abstract addresses my previous concerns, but note the old abstract is still being shown as the abstract at the head of the manuscript — this needs to be replaced in the peerj manuscript submission system.

ln 118-138 — appropriately toned down from the original. However, I still don't think the current manuscript adequately addresses the health benefits of lactobacilli and the current language seems to imply a lack of scientific understanding of the health benefits conferred by some lactobacilli, which simply isn't true. Much is known about the mechanisms by which SOME strains confer these benefits, but these cannot be generalized as they are not universal across species or strains of LAB.

Fig 1 - why only chicha? Why not show the water and unfermented samples? Also, BLAST simply is not a reliable taxonomic assigner for species-level discrimination of LAB using such a short amplicon. Please show the RDP classifications, not BLAST.

Fig 2 - This still is not justifiable — these short amplicon lengths are not adequate for reliable species-level discrimination of LAB. BLAST is misleading since it chooses the top hit without considering the confidence of this assignment. Furthermore, you are assuming that just because the top hit is a species that has some probiotic potential that all strains of that species are automatically probiotic. This figure should be removed as it is misleading, uses poor methodology, and is redundant with Fig 1.

ln 311-312 — interact with the human microbiome? Indeed, these are being consumed, but you have not looked at human microbiota samples so do not know whether any interactions occur. This statement should be removed.

ln 312-315 — This is an extremely speculative statement/figure and a little over the top, given the data that you have. You still do not know where these LAB species are coming from, that there is any exchange with the human microbiome, and that cyclic co-evolution occurs by continual passage through a human host. A certain amount of speculation is always healthy, but this is exaggerated, steps too far from the available data, and is misleading. I agree that ln 316-323 give a more moderated level of speculation that is important to this discussion.

ln 324-325 — You did not detect reliable species-level OTUs, so it is inappropriate to make this link based on BLAST assignments. I recommend removing this sentence.

---

## Round 0.3 · accepted · Accept

You responded well to the reviewer's comments, even though I would like to stress that the fact that others have used a certain method doesn't make it right. I still agree with the reviewer that Blast analysis of short read sequences doesn't make species level assignment better only because it gives a result.